# ADVANCING TEST-TIME ADAPTATION FOR ACOUSTIC FOUNDATION MODELS IN OPEN-WORLD SHIFTS

## ABSTRACT

Test-Time Adaptation (TTA) is a critical paradigm for tackling distribution shifts during inference, especially in visual recognition tasks. However, while acoustic models face similar challenges due to distribution shifts in test-time speech, TTA techniques specifically designed for acoustic modeling in the context of open-world data shifts remain scarce. This gap is further exacerbated when considering the unique characteristics of acoustic foundation models: 1) they are primarily built on transformer architectures with layer normalization and 2) they deal with test-time speech data of varying lengths in a non-stationary manner. These aspects make the direct application of vision-focused TTA methods, which are mostly reliant on batch normalization and assume independent samples, infeasible. In this paper, we delve into TTA for pre-trained acoustic models facing open-world data shifts. We find that noisy, high-entropy speech frames, often non-silent, carry key semantic content. Traditional TTA methods might inadvertently filter out this information using potentially flawed heuristics. In response, we introduce a learning-based adaptation enriched by confidence enhancement. Noting that speech signals' short-term consistency, we also apply consistency regularization during test-time optimization. Our experiments on synthetic and real-world datasets affirm our method's superiority over existing baselines.

## 1 INTRODUCTION

Deep neural networks (DNNs) have exhibited remarkable performance in scenarios where the training and testing sets adhere to the independent and identically distributed (i.i.d) assumption. However, real-world applications frequently involve domain shifts between the training to testing sets, such as visual variations due to evolving weather conditions in vision tasks (Hendrycks & Dietterich, 2019; Koh et al., 2021) and variations in timbre due to changing speakers in speech-related tasks (Liao, 2013). Unfortunately, DNNs are susceptible to performance degradation under such domain shifts, underscoring the importance of adapting DNN-based models to enhance their robustness in the face of open-world distribution shifts.

Test-Time Adaptation (TTA) emerges as a critical paradigm for addressing distribution shifts at inference time, which involves two lines of research, Test-Time Training (Sun et al., 2020) (TTT) and fully TTA (Wang et al., 2020). TTT necessitates more backward passes and source data to alter training with additional self-supervised objectives while fully TTA enables online updates of neural networks on test data in a source-free way, thus requiring a lower computational cost compared to TTT. Recent investigations (Niu et al., 2023; Zhou et al., 2023) have delved into TTA under the context of open-world data shifts, a more practical consideration for real-world applications. Notwithstanding TTA's success in tackling various forms of corruption in vision recognition tasks (Zhang et al., 2022; Boudiaf et al., 2022), the development of TTA techniques tailored for acoustic modeling in the context of open-world data shifts remains scarce.

In the light of human auditory system's inherent adaptability to real-world speech, it exhibits resilience in the face of diverse forms of speech corruption. However, while recent pre-trained acoustic foundation models, such as Wav2vec2 (Baevski et al., 2020), with task-specific fine-tuning achieve excellent performances in tasks such as Automatic Speech Recognition (ASR), they exhibit notable performance degradation when confronted with open-world speech during test-time, as depicted in Figure 1. Consequently, there exists an emergent demand to adapt these acoustic foundation models

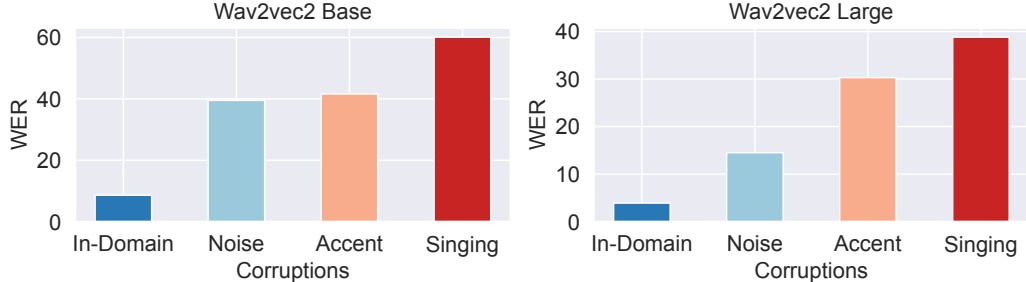

Figure 1: Robustness analysis of Wav2vec2 Base and Large on open-world corruptions including 1) Noise: additive noises on LibriSpeech test-other set, 2) Accent: accents of L2 learners on L2-Arctic subset 3) Singing: sung speech on DSing test set. In-Domain indicates the performance on LibriSpeech test-other set without additive noises. WER is short for Word Error Rate.

to open-world shifts when deployed in the real world. However, these models pose unique challenges due to their characteristics: 1) they are primarily built upon transformer architectures with layer normalization and 2) they deal with test-time speech data of varying lengths in a non-stationary manner. These distinctive features make it impractical to directly apply existing vision-focused TTA methods. These TTA techniques heavily rely on batch normalization, which acoustic foundation models lack. Additionally, they assume sample independence, an assumption that does not hold in the context of speech data.

In this work, we investigate the TTA of pre-trained foundation models facing open-world data shifts. Specifically, we focus on fully TTA to avoid altering the training of acoustic foundation models. Our goal is to leverage publicly available pre-trained acoustic models and adapt them to open-world data shifts. We initially follow the heuristic-based TTAs from prior works, such as Niu et al. (2023) designed for image classifications, to pinpoint a substantial proportion of noisy frames within non-silent speech segments before adaptation. Although Niu et al. (2023) characterized these high-entropy noisy frames as unreliable and potentially harmful for model adaptation, we observed that merely discarding these noisy non-silent frames adversely affected model performance. This is because these frames contain vital semantic information crucial for accurate recognition. Consequently, rather than excluding these frames, we introduce a learning-driven method, termed Confidence Enhanced Adaptation (CEA), designed to 'denoising' the intermediate representation of these noisy frames.

Additionally, we emphasize that frames within a short speech segment are temporally coherent, largely due to the consistent nature of phonemic content within such windows. This contrasts with image samples in a batch, which are frequently treated as independent entities. We conduct a wide range of experiments on both synthetic and real-world datasets, systematically assessing the model's robustness against Gaussian noises, environmental sounds, accents of second language (L2) learners, and singing (a.k.a sung speech). The experimental results substantiate the superiority of our proposed method over existing baselines.

In summary, our contributions are summarized as follows:

- We conduct an analysis of the robustness of acoustic foundation models under open-world speech data shifts, revealing that noisy speech frames with high entropy are frequently non-silent and bear critical semantic content.

- We introduce a learning-based adaptation approach enriched by confidence enhancement to boost the reliability of noisy frames and apply short-term consistency regularization for acoustic foundation models at test-time adaptation.

- We perform a wide range of experiments on both synthetic and real-world datasets, including novel experiments on real-world music datasets for the first time, thus contributing to the TTA community. Empirical results substantiate the superior performance of our method over existing baselines.

## 2 RELATED WORK

**Test-time Adaptation**. Test-time adaption plays an essential role in addressing distribution shifts encountered in test samples for a given pre-trained source model. Existing TTA methods can be categorized into two primary approaches: test-time training (TTT) (Sun et al., 2020) and fully TTA (Wang et al., 2020). TTT methods commonly incorporate additional self-supervised objectives during the model training phase (Liu et al., 2021; Bartler et al., 2022). In contrast, fully TTA exclusively updates models during the test phase using unsupervised objectives. Notably, fully TTA methods in the domain of computer vision have relied on Batch Normalization layers (Ioffe & Szegedy, 2015; Lim et al., 2023; Niu et al., 2022) while recent works (Niu et al., 2023) have begun to explore the potential of transformer-based models such as Vision Transformer (ViT) (Dosovitskiy et al., 2020), which employs layer normalization (Ba et al., 2016). Furthermore, there has been a growing interest in configuring TTA methods to suit real-world deployment scenarios that involve dynamic changes in environmental conditions (Wang et al., 2022). While vision-centric TTA approaches (Wang et al., 2022; Gong et al., 2022) exhibit an ability to address non-i.i.d data streams in fluctuating environments, they continue to operate under the assumption of sample independence within the same batch, rendering them less applicable to speech data. Despite the plethora of TTA methods, real-world data shifts encompassing both covariate and label shifts pose challenges to real-world deployment (Koh et al., 2021; Niu et al., 2023; Zhou et al., 2023). Consequently, further investigation is needed to address these challenges, and this paper focuses on tackling them.

**Robustness in Speech**. The realm of robust speech processing has a rich historical backdrop (Abdel-Hamid et al., 2012; Li et al., 2014; Kim & Stern, 2016). Prior studies have explored the acoustic shifts with a focus on distinct aspects, such as speaker adaptation (Liao, 2013), and accent adaptation (Yang et al., 2023b), often treating these facets in isolation. Consequently, these approaches encounter challenges when confronted with the broader context of open-world data shifts. Another research line focuses on the development of adaptation approaches for acoustic or speech models by reprogramming input data (Yang et al., 2021; 2023a;b) in a parameter-efficient manner, or designing wave prompts (Gao et al., 2022). A notable distinction between these works and TTA is their reliance on labeled target data pairs for supervised learning, as opposed to unsupervised TTA. Furthermore, despite the recent success of the large pre-trained acoustic model, the development of TTA methods for such acoustic foundation models remains scarce. Recent work (Lin et al., 2022; Kim et al., 2023) provides a pilot study on TTA for ASR and demonstrates the effectiveness of existing entropy minimization in the new setting. Our work focuses on designing generic TTA methods for pre-trained acoustic foundation models under open-world speech data shifts.

## 3 PRELIMINARY

We center our focus on the fully Test-Time Adaptation framework, characterized by episodic model adaptation, where the model is reset after processing each utterance. We denote the pre-trained acoustic foundation model as $f_\Theta(y|x)$. We investigate the core parts shared by most acoustic foundation models such as Wav2vec2 (Baevski et al., 2020), HuBERT (Hsu et al., 2021), WavLM (Chen et al., 2022) and Whisper (Radford et al., 2023), which can be typically decomposed into two constituent components: a feature extractor $g_\phi(z|x)$, parameterized by $\phi$, and a transformer encoder $h_\theta(y|z)$, parameterized by $\theta$. This decomposition is expressed as:

$$f_\Theta(y|x) = h_\theta(g_\phi(x)) \tag{1}$$

where $\Theta = \{\theta, \phi\}$ represents the collective set of model parameters. The feature extractor $g_\phi$ takes as input waveform audio or log-mel spectrogram. The transformer encoder $h_\theta$ serves as an audio encoder and outputs acoustic representations. Considering a test-time speech sequence $x_{1:n}$ of variable length $n$, typically with arbitrary domain shifts, the primary objective entails adapting the pre-trained acoustic model $f_\Theta$ to enhance its performance for $x_{1:n}$.

## 4 METHOD

In this section, we first analyze the common source of open-world shifts in the speech domain, and then provide our findings and methods for addressing open-world shifts. The overview of our method is presented in Figure 2.

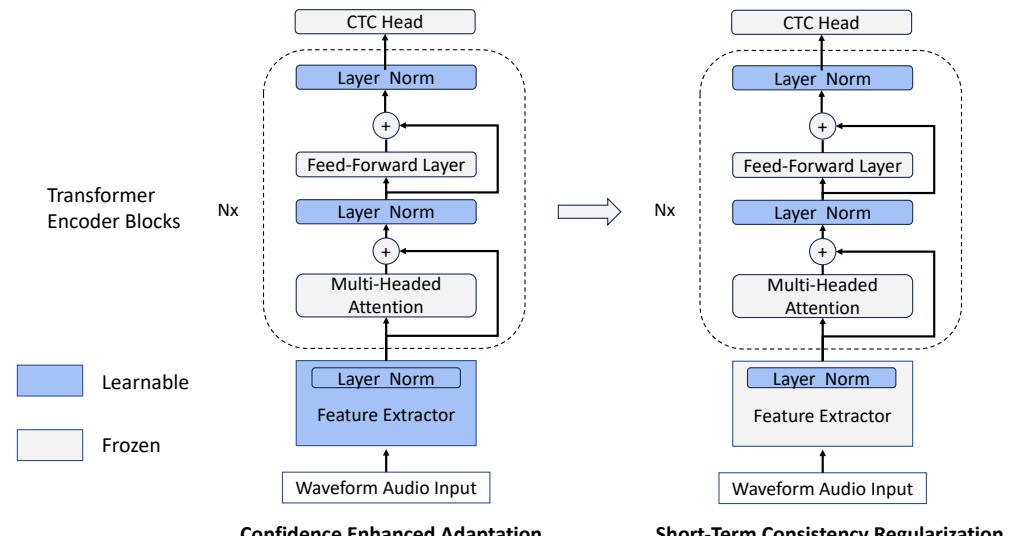

Figure 2: The overall framework of the proposed method. The figure takes a Connectionist Temporal Classification (CTC) based acoustic foundation model as an example. This framework involves two steps. The confidence enhanced adaptation is first performed to boost the reliability of noisy frames. The temporal consistency regularization is employed across the entire input sequence and jointly optimized with entropy minimization.

## 4.1 OPEN-WORLD SHIFTS IN THE SPEECH DOMAIN

Open-world distribution shifts encountered within the speech domain may originate from several sources, including:

**Speaker Changes**. Timbre variations in speech stemming from changes in the speaker's identity.

**Environmental Noises**. Perturbations introduced by ambient noises in the recording environments.

**Pronunciation Changes**. Alteration in pronunciation characteristics such as accent or singing.

**Text-domain Changes**. Shifts in the linguistic content or context of the speech data.

It is noteworthy that speaker changes, environmental noises, and pronunciation changes are typically categorized as covariate shift, as they pertain to variations in the input data distribution. In contrast, text-domain changes are categorized as label shift, as they involve alterations in the output distribution. Furthermore, it is important to acknowledge that real-world speech data often exhibit shifts stemming from multiple sources simultaneously, rendering the task of adaptation to open-world shifts complex and challenging.

## 4.2 CONFIDENCE ENHANCED ADAPTATION

To gain insights into the behavior of pre-trained acoustic models at the frame-level prediction in the presence of open-world distribution shifts, our initial analysis centers on the entropy distribution of speech data subjected to such shifts. We conducted experiments using both the LibriSpeech test-other dataset, which was deliberately corrupted by additive Gaussian noises, and the DSing test set. These experiments were performed with the Wav2vec2 Base model. We subsequently evaluated the percentages of high-entropy and low-entropy frames for both non-silent and silent speech segments. The classification of frames as silent or non-silent was determined based on pseudo labels derived from model predictions.

As illustrated in Figure 3, our findings reveal that, prior to any adaptation (Step=0), within the non-silent frames category, there exists a prevalence of high-entropy frames compared to low-entropy ones for Base models. Conversely, the opposite trend is observed within the silent frames category.

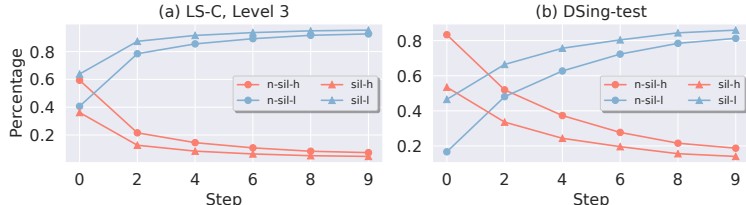

Figure 3: Distribution of Entropy in Acoustic Models: the entropy distributions are computed for Wav2vec2 Base models on the LibriSpeech noise-corrupted test-other and DSing test datasets across adaptation steps. We employ a threshold of $0.4 * \ln C$, as recommended in Niu et al. (2022), where $C$ represents the number of task classes. Frames with entropy values exceeding this threshold are highlighted in red, indicating high-entropy (h) frames, while low-entropy (l) frames are marked in blue. We use $\bullet$ to denote non-silent (non-sil) frames and $\triangle$ for silent (sil) frames. The training steps range from 0 to 9, and the results presented in each subfigure are based on the average of 100 random samples.

It is worth noting that existing literature provides heuristic insights suggesting that high-entropy samples may be unreliable and could potentially have a detrimental impact on model adaptation. However, it is crucial to recognize that these noisy frames contain essential content information that is critical for downstream tasks such as speech recognition. While prior research suggests that filtering out such unreliable samples may aid in stabilizing adaptation and improving performance, this approach proves infeasible in our specific case.

In response, we have proposed a learning-based adaptation approach aimed at enhancing the confidence of uncertain predictions, particularly for noisy frames. Denoting $\hat{y}_i^c = f_\Theta(c|x_{1:n})$ as the predicted probability of class $c$ for $i$-th frame, we quantify uncertainty through entropy, defined as:

$$E(x_i) = -\sum_c \hat{y}_i^c \log \hat{y}_i^c \tag{2}$$

Traditional heuristic-based Test-Time Adaptation (TTA) often relies on manually set thresholds for filtering our data samples of high entropy. In contrast, our approach utilizes pseudo labels $\hat{y}_i$ assigned to each frame $x_i$ and applies entropy minimization exclusively to non-silent frames, without the need of setting such thresholds. Specifically, we minimize a weighted entropy function, expressed as follows:

$$\min_{\Theta'=\{\phi,\theta_{LN}\}} \sum_{i=1}^n S(x_i)E(x_i), \tag{3}$$

where $\theta_{LN}$ denotes the affine parameters associated with layer normalization in the transformer encoder $h$, and $S(x_i)$ represents frame-level weights, defined as:

$$S(x_i) = \frac{1}{1 + \exp(-E(x_i))} \mathbb{I}_{\hat{y}i \neq c_0}(x_i), \tag{4}$$

where $c_0$ signifies the index corresponding to silent frames, and $\mathbb{I}$ is an indicator function. Such design empowers the model to assign greater importance to frames where it exhibits lower confidence. The increased weight encourages the model to focus more on these uncertain frames during adaptation, potentially leading to heightened model confidence on such frames. We term this approach "confidence-enhanced adaptation". Notice that this adaptation process entails an update of the feature extractor $g_\phi$. This empowers models with the capability to adapt to open-world shifts, even in the presence of substantial covariate shifts. As evidenced in Figure 3, the count of high-entropy frames diminishes while low-entropy frame counts increase with each adaptation step, underscoring the effectiveness of confidence-enhanced adaptation.

### 4.3 SHORT-TERM CONSISTENCY OF SPEECH SIGNALS

In the domain of speech signal processing, a salient characteristic is the short-term stability, where successive speech frames often convey the same phoneme or speech unit. This intrinsic temporal correlation is a defining attribute of speech data. Nevertheless, conventional Test-Time Adaptation (TTA) methods largely overlook this inherent temporal correlation within individual speech sequences.

To address this limitation, we propose a feature-wise short-term consistency regularization technique. We perform this regularization step after the confidence-enhanced adaptation process. This sequencing is deliberate as introducing temporal regularization over representations of noisy frames can potentially confuse models and yield undesirable optimization outcomes. Concretely, the regularization is jointly optimized alongside entropy minimization, as represented by the following equation:

$$\min_{\Theta_{LN}} \sum_{i=1}^{n} E(x_i) + \alpha \sum_{i=1}^{n-k+1} ||z'_{k+i-1} - z'_i||_2 \mathbb{I}_{\hat{y}_i \neq c_0}(x_i), \tag{5}$$

where $\alpha$ denotes the weight assigned to the regularization loss, and $\Theta_{LN}$ represents the affine parameters associated with layer normalization across the entire acoustic model. Here, $z_i$ signifies the feature representation of $i$-th frame obtained from the fine-tuned feature extractor, and $z'_i$ represents the modified feature representation achieved through a parameter-free self-attention operation. The parameter $k$ denotes the size of the window considered as the neighborhood of frame $x_i$. This regularization technique effectively captures the inherent temporal consistency found in speech data by compelling the representation of $x_i$ to closely resemble that of its neighboring frames within a predefined window.

## 5 EXPERIMENTS

In this section, we undertake an evaluation of the robustness of acoustic foundation models against various forms of open-world corruption. We discuss the robustness against synthetic noises including Gaussian noises and real-world environmental sounds in Section 5.2, real-world data shifts including L2 accents and singing voice (sung speech) in Section 5.3, and decoding strategy pertaining to language models in Section 5.4.

### 5.1 EXPERIMENTAL SETUP

**Datasets**. Our experiments involve the utilization of four distinct datasets: two synthetic and two real-world datasets. The first synthetic dataset, named LS-C, represents the LibriSpeech (Panayotov et al., 2015) test-other set Corrupted by additive Gaussian noises. We introduce five levels of severity to simulate various degrees of corruption as per Hendrycks & Dietterich (2019) for evaluating the trend of model robustness. Higher levels indicate more severe corruption although heavily corrupted speech data may not be common cases in the real world. Subsequently, the second synthetic dataset, named LS-P, is the LibriSpeech test-other set Perturbed by real-world environmental sounds. This dataset encompasses eight diverse types of environmental sound, including Air Conditioner, Babble, Munching, Shutting Door, Vacuum Cleaner, Airport Announcements, Copy Machine, and Typing. These environmental sounds are from the MS-SNSD noise test set (Reddy et al., 2019). Each type is added to the original audio with five distinctive signal-to-noise ratios (SNRs) representing five levels of severity. Our study further extends to two real-world datasets with open-world data shifts. The L2-Arctic (Zhao et al., 2018) dataset comprises speech data from second language (L2) learners originating from six countries with different first languages (L1): Arabic, Mandarin, Hindi, Korean, Spanish, and Vietnamese. Furthermore, we broaden our investigation to encompass music datasets, DSing (Dabike & Barker, 2019) and Hansen (Hansen & Fraunhofer, 2012), featuring singing voice (sung speech). More details of dataset statistics can be found in Appendix A.1 and details of implementation can be found in Appendix A.2.

**Baselines**. To assess the adaptation performance of our proposed method, we consider the following TTA baselines. **Tent** (Wang et al., 2020) adapt transformation layers with the objective of entropy

minimization. Despite it being initially proposed for batch normalization, we refer to updating the affine parameters of layer normalization as Tent in our work. In addition, we involve the baseline **TeCo** (Yi et al., 2023), originally proposed for video classification with temporal coherence regularization, due to its applicability to sequential data. Our comparison also includes the **SAR** (Niu et al., 2023), specifically designed to address data shifts in the dynamic wild world. Furthermore, we also introduce comparisons with **SUTA** (Lin et al., 2022) using entropy minimization and minimum class confusion, and **SGEM** (Kim et al., 2023) using sequential-level generalized entropy minimization in conjunction with beam search employing language models.

## 5.2 ROBUSTNESS TO SYNTHETIC NOISE

### 5.2.1 GAUSSIAN NOISES

In the initial phase of our experiments, we focus on synthetic data and assess the robustness in the presence of various levels of Gaussian noise injected into the test speech audio. The outcomes are reported in Table 1. It is observed that our proposed method consistently outperforms existing baseline approaches across five levels of noise. Notably, our approach achieves a relative improvement of 32.0% on average in terms of WER, when compared to using the source model without adaptation.

Furthermore, it is imperative to note that SAR, designed for addressing data shifts in dynamic real-world scenarios, demonstrates comparatively less improvement compared with the Tent method. This observation underscores the limitations of filtering noisy frames for speech recognition. Instead, the learning-based adaptation adopted in our method shows superiority. Moreover, we discover that TeCo provides marginal improvement compared to Tent, indicating that coherence regularization is limited in the context of noisy frames. In contrast, our confidence-enhanced adaptation yields further benefits for temporal consistency regularization.

| Method | Level 1 | Level 2 | Level 3 | Level 4 | Level 5 | Average |
|--------|---------|---------|---------|---------|---------|---------|
| Source | 13.9 | 24.4 | 39.5 | 54.5 | 75.7 | 41.6 |
| Tent | 11.6 | 19.7 | 32.2 | 46.3 | 69.2 | 35.8 |
| SAR | 12.7 | 21.5 | 35.0 | 49.2 | 72.0 | 38.1 |
| TeCo | 13.6 | 19.7 | 32.2 | 46.3 | 69.3 | 35.8 |
| SUTA | 10.9 | 16.7 | 24.6 | 34.7 | **56.5** | 28.7 |
| Ours | **10.7** | **16.2** | **24.0** | **34.1** | 56.5 | **28.3** |

Table 1: WER (%) results on LS-C over five severity levels of Gaussian noises using Wav2vec2 Base with greedy decoding. The best results are bold.

### 5.2.2 ENVIRONMENTAL SOUNDS

We further evaluate the robustness on LS-P, which introduces eight common environmental sounds in the test audio at five levels of severity. The results of adding Air Conditioner sound and Typing sound are reported in Table 2 and Table 3 respectively (Full experimental results can be found in Appendix A.3). It is noticeable that our method can yield over 30% relative improvements in low-SNR scenarios. Notably, for the case with 5 dB SNR in Table 2, our method demonstrates a substantial 41.7% relative improvement, suggesting its efficacy in mitigating the impact of real-world environmental sound corruption.

## 5.3 ROBUSTNESS TO REAL-WORLD DATA SHIFTS

### 5.3.1 L2 ACCENTS

Data shifts resulting from accent variations are a common occurrence in real-world scenarios, arising from differences in dialects or non-native speech patterns. Another pertinent instance of such shifts is encountered in children's speech, which is also a common pronunciation change and one type of accent in the real world. In order to assess the robustness to such pronunciation variations, we undertake the test-time adaptation to accents exhibited by L2 learners using the L2-Arctic dataset.

|        | 10   | 5    | 0    | -5   | -10  |
|--------|------|------|------|------|------|
| Source | 28.1 | 43.9 | 65.0 | 83.4 | 94.2 |
| Tent   | 22.6 | 36.1 | 56.6 | 77.9 | 91.4 |
| SAR    | 24.5 | 39.1 | 59.9 | 79.9 | 92.1 |
| TeCo   | 22.5 | 36.2 | 56.6 | 77.9 | 91.3 |
| SUTA   | 17.7 | 26.1 | 41.2 | 62.7 | 82.7 |
| Ours   | **17.5** | **25.6** | **40.6** | **61.6** | **82.2** |

Table 2: WER (%) results on **Air Conditioner** sound over five severity levels using Wav2vec2 Base with greedy decoding. SNRs (dB) are listed in the first row. The best results are bold.

|        | 10   | 5    | 0    | -5   | -10  |
|--------|------|------|------|------|------|
| Source | 26.2 | 34.0 | 44.4 | 56.4 | 69.0 |
| Tent   | 21.0 | 27.9 | 37.0 | 49.2 | 63.0 |
| SAR    | 23.0 | 30.3 | 39.7 | 52.1 | 65.3 |
| TeCo   | 21.0 | 27.8 | 37.0 | 49.1 | 63.0 |
| SUTA   | 17.9 | 23.3 | 30.4 | 41.0 | 53.4 |
| Ours   | **17.5** | **22.8** | **29.9** | **40.4** | **52.6** |

Table 3: WER (%) results on **Typing** sound over five severity levels using Wav2vec2 Base with greedy decoding. SNRs (dB) are listed in the first row. The best results are bold.

To comprehensively evaluate the performance, we evaluate all speakers for each L1 and present the speaker-level results for each L1 in Appendix A.4. The experimental findings consistently underscore the superiority of our proposed method across different L1 categories.

### 5.3.2 Singing Voice

In this session, We discuss the robustness of speech models to singing voice for the first time. Singing, also referred to as sung speech, is characterized by a distinctive pronunciation pattern. Notably, it encompasses various frequency fluctuations, including the apparent pitch variations along with the melody. This constitutes a tremendous covariate shift, rendering the adaptation from speech to singing more challenging than that from speech to speech. Moreover, the existence of professional singing techniques further compounds the challenges associated with adaptation. For instance, the elongation of word pronunciation, a common occurrence in singing, is a departure from typical speech patterns.

To evaluate the adaptation performance under shifts from singing voice, we conduct experiments on three music datasets, utilizing both Wav2vec2 Base and Wav2vec2 Large models. The outcomes are presented in Table 4. The results indicate that our proposed method consistently attains the best performances for both Base and Large models. In addition, the Wav2vec2 Large model exhibits superior robustness than the Base model. Nevertheless, it still experiences a noticeable performance degradation when compared with adaptation in noise and accent robustness evaluations, suggesting the limited ability of acoustic foundation models under huge real-world data shifts.

| Method | DSing-dev | | DSing-test | | Hansen | | Average | |
|--------|------|-------|------|-------|------|-------|------|-------|
|        | Base | Large | Base | Large | Base | Large | Base | Large |
| Greedy Search | | | | | | | | |
| Source | 61.8 | 40.6 | 60.1 | 38.8 | 64.3 | 43.7 | 62.1 | 41.0 |
| Tent   | 55.7 | 34.8 | 56.1 | 33.2 | 60.2 | 39.1 | 57.3 | 35.7 |
| SAR    | 58.8 | 40.6 | 57.2 | 38.2 | 62.7 | 42.7 | 59.6 | 40.5 |
| TeCo   | 56.2 | 35.0 | 55.6 | 33.1 | 60.0 | 39.1 | 57.3 | 35.7 |
| SUTA   | 53.9 | 34.9 | 51.3 | 33.6 | **58.0** | 39.3 | 54.4 | 35.9 |
| Ours   | **53.5** | **34.0** | **50.1** | **31.2** | **58.0** | **37.9** | **53.9** | **34.4** |
| Beam Search | | | | | | | | |
| Source+LM | 58.6 | 41.1 | 55.3 | 37.6 | 60.1 | 43.5 | 58.0 | 40.7 |
| SGEM   | 54.4 | 34.4 | 50.8 | 33.0 | 57.8 | 38.6 | 54.3 | 35.3 |
| Ours+LM | **53.2** | **33.3** | **50.0** | **30.3** | **57.7** | **37.5** | **53.6** | **33.7** |

Table 4: WER (%) results on DSing-dev, DSing-test, and Hansen with greedy search and beam search. Base and Large denote Wav2vec2 Base and Wav2vec2 Large respectively. The best results are bold.

| Method | DSing-dev | Dsing-test |
|---|---|---|
| Ours | **53.5** | **50.1** |
| w/o STCR | 54.4 | 51.0 |
| w/o CEA | 55.7 | 54.5 |

Table 5: Ablation study of core components proposed in our work. WER (%) results are reported.

| Strategy | DSing-dev | Dsing-test |
|---|---|---|
| Non-Silent | **53.5** | **50.1** |
| Silent | 54.9 | 51.7 |
| All | 54.9 | 50.6 |

Table 6: Ablation study of strategies for frame selection. WER (%) results are reported.

## 5.4 DECODING STRATEGIES

We discuss the decoding strategies employed in experiments in this session. In our preceding experiments, we mainly utilize greedy decoding, which does not explicitly tackle the text-domain changes. In the subsequent analysis, we compare our proposed method with SGEM, which leverages beam search for decoding. The results are presented in Table 4. Notably, our findings reveal that even in the absence of explicit adaptation for the language model, our approach still consistently outperforms SGEM. We also observe that the results achieved by our method using greedy search can, on average, surpass those of SGEM. We conjecture that our proposed short-term consistency regularization addresses the label shift implicitly by fostering label coherency among neighbor frames. Moreover, it is discovered that the enhancements facilitated by adaptation are more pronounced compared to the ones achieved through beam search, indicating the significance of test-time adaptation for acoustic foundation models.

## 6 ABLATION STUDY

**Effects of Components**. We conduct the ablation study on DSing-dev and DSing-test using Wav2vec2 Base with greedy search to dissect the individual impact of two core components proposed in our methods. The results presented in Table 5 illustrate that the removal of short-term consistency regularization (STCR) leads to a relatively modest decline in performance, in contrast to the more substantial deterioration observed upon the removal of confidence enhanced adaptation (CEA). This observation underscores the significance of our proposed CEA. Furthermore, the introduction of STCR yields additional performance gains when employed in conjunction with CEA.

**Comparison with strategies for frame selection**. We proceed to analyze strategies utilized for the selection of speech frames optimized within the CEA framework. We investigate three pseudo-label-based strategies, namely a) selection of non-silent frames (as used in our method), b) selection of silent frames, and c) selection of all frames. The results are detailed in Table 6. The empirical findings reveal that the optimization of silent frames or all frames within CEA yields inferior performance compared to the optimization of non-silent frames. Moreover, it is observed that the degradation is not so substantial, as optimizing silent or all frames may also contribute to enhancing the reliability of noisy frames.

## 7 CONCLUSIONS

In this paper, we study the fully Test-Time Adaptation of pre-trained acoustic foundation models to address open-world data shifts. By investigating the role of noisy frames with high entropy within non-silent speech segments, we introduce a novel Confidence Enhanced Adaptation method to enhance the reliability of noisy frames via denoising their intermediate representations rather than discarding them. Moreover, our emphasis on short-term consistency of speech signals leads us to apply consistency regularization, yielding further improvement in WER performance for speech data. Extensive experiments on synthetic and real-world datasets demonstrated the effectiveness of our approach over existing baselines under the open-world data shifts. However, it remains challenging to adapt language models to address text-domain shifts due to the unavailability of target domain texts in the TTA setting. Consequently, we consider incorporating large language foundation models into the recognition decoding process as a promising direction in future work for tackling open-world text-domain shifts.

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

# A   APPENDIX

## A.1   DATASET DETAILS

We show the statistics of datasets used in our work in Table 7 where # Utt. indicates the total number of utterances. We build our synthetic datasets on LibriSpeech test-other set. For LS-C, we add the Gaussian noises when preparing the data loader and use the amplitudes {0.005, 0.01, 0.015, 0.02, 0.03} as level 1-5 severity. For LS-P, we use the AirConditioner_6, Typing_2, Babble_4, Munching_3, ShuttingDoor_6, VacuumCleaner_1, AirportAnnouncements_2, CopyMachine_2 wave files from MS-SNSD [1] as the environmental sounds and synthesize audios with signal-to-noise ratios {10, 5, 0, -5, -10} seperately. For L2-Arctic, we use the default splits of 24 non-native speakers with a balanced gender and L1 distribution. For music datasets, we use the default DSing dev and test sets and the full Hansen set (no split).

| Type | Datasets | # Utt. | Total Duration |
|---|---|---|---|
| Noise | LS-C | $2939 \times 5$ | $5.1 \times 5$ h |
| | LS-P | $2939 \times 8 \times 5$ | $5.1 \times 8 \times 5$ h |
| Accent | L2-Arctic | 26867 | 27.1 h |
| Music | DSing-dev | 482 | 41 min |
| | DSing-test | 480 | 48 min |
| | Hansen | 634 | 34 min |

Table 7: Statistics of evaluation datasets.

## A.2   IMPLEMENTATION DETAILS

In our experimental evaluations, we mainly employ the acoustic foundation model, Wav2vec2. Specifically, we utilize its Connectionist Temporal Classification (CTC) variants with different model sizes, Wav2vec2 Base and Wav2vec2 Large. We involve the usage of publicly available Wav2vev2 Base [2] and Wav2vec2 Large [3] models fine-tuned on speech recognition tasks. We mainly conduct experiments on these two models despite the applicability of our method to other transformer-based architectures of acoustic foundation models. To make a fair comparison with methods employing beam search, we utilize the same 4-gram language model [4] as SGEM. Since our test-time setting requires no access to the target text, we use the language model trained on the speech dataset despite the text-domain shift. All speech inputs are sampled or resampled at 16Khz.

We use Pytorch and Huggingface Transformers in our implementation. All experiments are run on a single NVIDIA A5000 GPU (24G). We evaluate the performance of all baselines after adaptation for ten steps. We use the AdamW optimizer as default for all experiments. The weight $\alpha$ of consistency regularization is set to be 0.3. We consider the learning rate in {2e-4, 5e-4, 8e-4} for tuning affine parameters of layer normalization and consider the learning rate in {2e-5, 5e-5} for tuning feature extractor.

## A.3   FULL RESULTS FOR LS-P

We present the full WER results for eight environmental sounds of five severity levels in Table 8 - 15. The first row denotes signal-to-noise ratios.

## A.4   FULL RESULTS FOR L2-ARCTIC

We present the full speaker-level WER results for each L1 in Table 16 - 21. The first row denotes the speaker ID. The details of the speaker ID can be found in the L2-Arctic [5]

---

[1] https://github.com/microsoft/MS-SNSD

[2] https://huggingface.co/facebook/wav2vec2-base-960h

[3] https://huggingface.co/facebook/wav2vec2-large-960h-lv60-self

[4] https://huggingface.co/patrickvonplaten/wav2vec2-base-100h-with-lm

[5] https://psi.engr.tamu.edu/l2-arctic-corpus/

## A.5 RESULTS ON OTHER ACOUSTIC FOUNDATION MODELS

| | Size | Level 1 | Level 2 | Level 3 | Level 4 | Level 5 | Avg |
|---|---|---|---|---|---|---|---|
| **Wav2vec2** | | | | | | | |
| Source | Base | 13.9 | 24.4 | 39.5 | 54.5 | 75.7 | 41.6 |
| | Large | 5.0 | 8.1 | 14.6 | 24.9 | 46.9 | 19.9 |
| Ours | Base | 10.7 | 16.2 | 24.0 | 34.1 | 56.5 | 28.3 |
| | Large | 4.3 | 6.1 | 9.7 | 15.1 | 31.1 | 13.3 |
| WERR (%) | Base | 23.0 | 33.6 | 39.2 | 37.4 | 25.4 | 31.7 |
| | Large | 14.0 | 24.7 | 33.6 | 39.4 | 33.7 | 29.1 |
| **Hubert** | | | | | | | |
| Source | Base | 26.1 | 32.7 | 40.6 | 49.0 | 63.4 | 42.4 |
| | Large | 5.0 | 6.4 | 8.9 | 12.8 | 24.3 | 11.5 |
| Ours | Base | 19.3 | 23.7 | 28.9 | 35.0 | 47.5 | 30.9 |
| | Large | 4.3 | 5.2 | 6.9 | 9.1 | 16.1 | 8.3 |
| WERR (%) | Base | 26.1 | 27.5 | 28.8 | 28.6 | 25.1 | 27.2 |
| | Large | 14.0 | 18.8 | 22.5 | 28.9 | 33.7 | 23.6 |
| **WavLM** | | | | | | | |
| Source | Base | 24.1 | 35.9 | 48.2 | 59.8 | 76.7 | 48.9 |
| | Large | 14.4 | 17.5 | 21.5 | 26.1 | 36.1 | 23.1 |
| Ours | Base | 15.1 | 19.8 | 25.9 | 32.8 | 47.6 | 28.2 |
| | Large | 10.7 | 12.4 | 14.5 | 17.1 | 23.9 | 15.7 |
| WERR (%) | Base | 37.3 | 44.8 | 46.3 | 45.2 | 37.9 | 42.3 |
| | Large | 25.7 | 29.1 | 32.6 | 34.5 | 33.8 | 31.1 |

Table 8: WER (%) results on LS-C over five severity levels of Gaussian noises using both base and large models of Wav2vec2, Hubert, WavLM with greedy decoding. WERR stands for word error rate reduction.

| | Wav2vec2 | | Hubert | | WavLM | |
|---|---|---|---|---|---|---|
| | Base | Large | Base | Large | Base | Large |
| Source | 60.1 | 38.8 | 71.5 | 43.9 | 76.1 | 66.2 |
| Ours | 50.1 | 31.2 | 62.4 | 32.4 | 59.6 | 51.1 |
| WERR (%) | 16.6 | 19.6 | 12.7 | 26.2 | 21.7 | 22.8 |

Table 9: WER (%) results on DSing-test using both base and large models of Wav2vec2, Hubert, WavLM with greedy decoding. WERR stands for word error rate reduction.

In an extension of the main experiments, we delved into the adaptation performance across diverse acoustic foundation models. Specifically, our additional experiments utilize various models including, Hubert-Base [6], Hubert-Large [7], WavLM-Base [8], and WavLM-Large [9] from Huggingface. These experiments are conducted to assess the adaptation performance in relation to different model sizes, and training data sources. The outcomes on the LS-C and DSing-test datasets are reported in Table 8

---

[6]https://huggingface.co/danieleV9H/hubert-base-libri-clean-ft100h

[7]https://huggingface.co/facebook/hubert-large-ls960-ft

[8]https://huggingface.co/patrickvonplaten/wavlm-libri-clean-100h-base-plus

[9]https://huggingface.co/patrickvonplaten/wavlm-libri-clean-100h-large

and Table 9 respectively. We employ the word error rate reduction (WERR) to measure the relative improvement brought by our adaptation method. We summarize the findings as follows:

**Model Sizes**. A comparative analysis is conducted between the base and large versions of each model. The findings reveal that large models consistently surpass base models. Furthermore, our proposed approach uniformly improves both base and large models. A notable observation is that our method elicits a greater average improvement in base models compared to large models within the LS-C dataset. This trend is particularly pronounced under lower noise levels ranging from 1 to 3. In contrast, within the DSing-test set, the enhancement for large models is more significant than for base models. The phenomenon may be attributed to the fact that large models already exhibit commendable performance under minor corruptions, even without adaptation, thus providing limited scope for further improvement. However, in scenarios involving significant shifts, the expansive parameterization of large models facilitates more effective adaptation, whereas base models face challenges.

**Training Data Sources**. A comparative evaluation of models trained with different datasets, including Wav2vec2-Large trained with 960h LibriSpeech set, Hubert-Large trained with 960h LibriSpeech set, and WavLM-Large trained with 100h LibriSpeech clean set, indicates that the larger-size data set establish a stronger foundation for test-time adaptation. A similar inference can be drawn when comparing Wav2vec2-Base trained with 960h LibriSpeech set, Hubert-Base trained with 100h LibriSpeech clean set, and WavLM-Base trained with 100h LibriSpeech clean set.

In summary, our proposed unsupervised TTA method demonstrates a considerable benefit across diverse acoustic foundation models, reflecting substantial improvements for different model sizes and training data sources.

|        | 10   | 5    | 0    | -5   | -10  |
|--------|------|------|------|------|------|
| Source | 28.1 | 43.9 | 65.0 | 83.4 | 94.2 |
| Tent   | 22.6 | 36.1 | 56.6 | 77.9 | 91.4 |
| SAR    | 24.5 | 39.1 | 59.9 | 79.9 | 92.1 |
| TeCo   | 22.5 | 36.2 | 56.6 | 77.9 | 91.3 |
| SUTA   | 17.7 | 26.1 | 41.2 | 62.7 | 82.7 |
| Ours   | **17.5** | **25.6** | **40.6** | **61.6** | **82.2** |

Table 10: Air Conditioner.

|        | 10   | 5    | 0    | -5   | -10  |
|--------|------|------|------|------|------|
| Source | 26.2 | 34.0 | 44.4 | 56.4 | 69.0 |
| Tent   | 21.0 | 27.9 | 37.0 | 49.2 | 63.0 |
| SAR    | 23.0 | 30.3 | 39.7 | 52.1 | 65.3 |
| TeCo   | 21.0 | 27.8 | 37.0 | 49.1 | 63.0 |
| SUTA   | 17.9 | 23.3 | 30.4 | 41.0 | 53.4 |
| Ours   | **17.5** | **22.8** | **29.9** | **40.4** | **52.6** |

Table 11: Typing.

|        | 10   | 5    | 0    | -5   | -10  |
|--------|------|------|------|------|------|
| Source | 50.4 | 62.8 | 74.6 | 83.8 | 90.1 |
| Tent   | 44.8 | 57.6 | 71.1 | 82.7 | 90.5 |
| SAR    | 47.3 | 57.8 | 72.1 | 82.5 | 89.6 |
| TeCo   | 44.8 | 57.6 | 71.1 | 82.7 | 90.5 |
| SUTA   | 39.7 | 51.9 | 64.4 | 76.4 | **85.2** |
| Ours   | **39.3** | **51.5** | **64.1** | **76.3** | 85.3 |

Table 12: Munching.

|        | 10   | 5    | 0    | -5   | -10  |
|--------|------|------|------|------|------|
| Source | 19.2 | 23.6 | 29.7 | 37.0 | 45.0 |
| Tent   | 16.4 | 20.5 | 26.0 | 33.0 | 41.5 |
| SAR    | 17.7 | 22.0 | 27.7 | 35.0 | 42.7 |
| TeCo   | 16.3 | 20.5 | 26.0 | 32.9 | 41.5 |
| SUTA   | 14.9 | 18.5 | 23.6 | 29.9 | 37.7 |
| Ours   | **14.8** | **18.3** | **23.4** | **29.7** | **37.4** |

Table 13: Shutting Door.

|        | 10   | 5    | 0    | -5   | -10  |
|--------|------|------|------|------|------|
| Source | 57.8 | 76.6 | 91.5 | 98.2 | 99.9 |
| Tent   | 49.7 | 69.2 | 87.2 | 97.0 | 99.6 |
| SAR    | 52.6 | 72.7 | 88.5 | 96.9 | 99.8 |
| TeCo   | 49.7 | 69.2 | 87.2 | 96.9 | 99.6 |
| SUTA   | 39.8 | 56.7 | 76.6 | 93.2 | **98.6** |
| Ours   | **39.3** | **56.0** | **76.0** | **93.0** | **98.6** |

Table 14: Vacuum Cleaner.

|        | 10   | 5    | 0    | -5   | -10  |
|--------|------|------|------|------|------|
| Source | 40.9 | 54.3 | 66.3 | 75.8 | 83.4 |
| Tent   | 36.1 | 49.3 | 62.8 | 73.7 | 82.4 |
| SAR    | 38.2 | 51.0 | 64.0 | 74.3 | 82.2 |
| TeCo   | 36.1 | 49.2 | 62.8 | 73.7 | 82.3 |
| SUTA   | **31.2** | 43.8 | 58.3 | **70.4** | **79.3** |
| Ours   | **31.2** | **43.7** | **58.1** | 70.5 | 79.7 |

Table 15: Airpoint Announcements.

|        | 10   | 5    | 0    | -5   | -10  |
|--------|------|------|------|------|------|
| Source | 49.8 | 63.5 | 76.6 | 86.9 | 93.5 |
| Tent   | 44.4 | 58.9 | 74.2 | 86.3 | 93.7 |
| SAR    | 46.6 | 60.7 | 74.8 | 86.2 | 93.2 |
| TeCo   | 44.4 | 58.8 | 74.2 | 86.2 | 93.7 |
| SUTA   | 39.3 | 52.7 | 67.4 | **80.8** | **89.7** |
| Ours   | **38.9** | **52.3** | **67.3** | 81.0 | 89.8 |

Table 16: Copy Machine.

|        | 10   | 5    | 0    | -5    | -10   |
|--------|------|------|------|-------|-------|
| Source | 66.6 | 81.6 | 94.7 | 104.3 | 111.2 |
| Tent   | 62.0 | 77.8 | 92.0 | 102.2 | 109.4 |
| SAR    | 62.8 | 77.7 | 90.5 | 102.1 | **106.9** |
| TeCo   | 61.9 | 77.8 | 91.9 | 102.2 | 109.4 |
| SUTA   | 55.5 | 73.0 | **88.6** | **101.1** | 109.2 |
| Ours   | **55.5** | **73.0** | 89.1 | 102.0 | 110.3 |

Table 17: Babble.

|        | ABA      | SKA      | YBAA     | ZHAA     |
|--------|----------|----------|----------|----------|
| Source | 21.0     | 32.5     | 16.7     | 17.3     |
| Tent   | 18.4     | 28.4     | 14.5     | 14.4     |
| SAR    | 19.4     | 30.3     | 15.7     | 15.3     |
| TeCo   | 18.4     | 28.4     | 14.5     | 14.4     |
| SUTA   | 17.8     | 27.2     | 13.7     | 14.0     |
| Ours   | **17.7** | **26.8** | **13.5** | **13.9** |

Table 18: Arabic.

|        | BWC      | LXC      | NCC      | TXHC     |
|--------|----------|----------|----------|----------|
| Source | 28.5     | 33.5     | 26.9     | 21.1     |
| Tent   | 24.1     | 29.2     | 22.8     | 18.1     |
| SAR    | 26.3     | 30.9     | 25.0     | 19.5     |
| TeCo   | 24.1     | 29.3     | 22.9     | 18.0     |
| SUTA   | 23.3     | **27.6** | 21.5     | 17.4     |
| Ours   | **23.0** | 27.7     | **21.3** | **17.3** |

Table 19: Mandarin.

|        | ASI      | RRBI     | SVBI     | TNI      |
|--------|----------|----------|----------|----------|
| Source | 14.3     | 15.7     | 19.8     | 18.6     |
| Tent   | 11.7     | 12.9     | 15.7     | 15.6     |
| SAR    | 12.7     | 14.0     | 17.6     | 16.7     |
| TeCo   | 11.7     | 13.0     | 15.8     | 15.6     |
| SUTA   | **11.3** | 12.5     | 14.3     | 14.9     |
| Ours   | **11.3** | **12.2** | **14.3** | **14.8** |

Table 20: Hindi.

|        | HJK     | HKK      | YDCK     | YKWK     |
|--------|---------|----------|----------|----------|
| Source | 11.8    | 23.3     | 17.2     | 17.0     |
| Tent   | 9.7     | 20.8     | 15.0     | 14.5     |
| SAR    | 10.9    | 21.7     | 15.8     | 15.5     |
| TeCo   | 9.8     | 20.8     | 15.0     | 14.5     |
| SUTA   | **9.5** | 19.8     | 14.2     | 13.8     |
| Ours   | **9.5** | **19.7** | **13.9** | **13.7** |

Table 21: Korean.

|        | EBVS     | ERMS     | MBMPS    | NJS      |
|--------|----------|----------|----------|----------|
| Source | 35.7     | 24.2     | 14.1     | 14.6     |
| Tent   | 31.7     | 20.0     | 12.7     | 12.4     |
| SAR    | 33.5     | 21.7     | 13.4     | 13.2     |
| TeCo   | 31.7     | 20.0     | 12.7     | 12.4     |
| SUTA   | 29.7     | 18.7     | **12.3** | **12.1** |
| Ours   | **29.5** | **18.5** | **12.3** | **12.1** |

Table 22: Spanish.

|        | HQTV     | PNV      | THV      | TLV      |
|--------|----------|----------|----------|----------|
| Source | 41.6     | 18.5     | 38.1     | 41.1     |
| Tent   | 38.0     | 16.4     | 34.4     | 38.1     |
| SAR    | 40.3     | 17.6     | 36.2     | 39.4     |
| TeCo   | 38.0     | 16.4     | 34.4     | 38.0     |
| SUTA   | 36.5     | **15.5** | 33.2     | **36.8** |
| Ours   | **36.3** | **15.5** | **32.9** | **36.8** |

Table 23: Vietnamese.

