# OpenReview forum: "Advancing Test-Time Adaptation for Acoustic Foundation Models in Open-World Shifts"
_ICLR.cc/2024/Conference — Submitted to ICLR 2024_

### Official Review · Reviewer_hWCQ · 2023-10-30

**Soundness:** 2 fair
**Presentation:** 2 fair
**Contribution:** 3 good
**Rating:** 6
**Confidence:** 5

**Summary:**

This paper aims to investigate test-time adaptation tasks for pre-trained acoustic and speech models, considering them foundational in their respective domains.

During test-time adaptation that involves frozen pre-trained models, the principal innovations of this work are encapsulated in two proposed methods: `Confidence Enhanced Adaptation` and `Short-Term Consistency Regularization.`

- Both methods release trainable parameters, whether through normalization layers or input-based feature modifications. These methods closely resemble the techniques used in neural speech model reprogramming, as reported in acoustic model reprogramming ICML 2021 [1], addressing similar challenges in the adaptation of frozen models.

Overall, the author seeks to address a critical issue in acoustic modeling and speech processing. However, the paper's effectiveness is compromised by a noticeable gap in the understanding of related work on frozen acoustic model adaptation. This lack, coupled with certain experimental setups and theoretical justifications, undermines the paper's ability to convincingly argue whether the proposed method is both novel and a parameter-efficient approach for the adaptation of frozen acoustic models.

As a reviewer, I will outline several potential and relevant perspectives for improvement below. Generally, I concur that the direction of open-set test-time adaptation warrants deeper study and is a worthwhile pursuit.

From a neutral standpoint, it is necessary for the authors to significantly revise the manuscript, as there are fundamental LaTeX errors that need correction. Currently, the draft falls short of the above-average quality expected of a paper submission.

For instance, the authors should use `\citep` for parenthetical references, `\citet` when referring to the author(s) as part of the narrative, and `\cite` when mentioning the work as a standalone noun. Further details on this can be found in the strengths and weaknesses sections.

**Strengths:**

- The topic of test-time adaptation is generally interesting to the ml / speech community, although some perspectives on foundation (very large pre-trained) models based on in-context learning are not covered.

**Weaknesses:**

- writing quality could be very improved (perhaps doing in their next version)
  - citations code / format usages
  - grammar issues
  - The literature review on frozen acoustic model-based adaptation lacks depth. The current review primarily applies existing algorithms from the ML community, and the proposed method inadvertently overlaps with reprogramming/prompting due to an inadequate literature survey.
    - for example, the additional losses based adapters / input-only training has been also proposed in [4]
- The theoretical connection between test-time adaptation efficacy and frozen pre-trained models is unclear. A well-known recent insight involves latent space alignment via Wasserstein measurements, yet the related discussion is absent here.
- The related speech processing references are missing
  - frozen model adaptation via Bayesian adaptation in speech
  - frozen model adaptation with trainable inputs reprogramming / prompting
- ablation experiments are missing on
  - Similar parameter-efficient baselines work on frozen pre-trained adaptation diagrams (e.g., input reprogramming, adapters, prompting), making it difficult to justify the novelty on applying trainable layer norms
  - In terms of frozen pre-trained models, only Wav2Vec 2.0 Large has been evaluated. The term "foundation model" is often used for models with over 1 billion or even 100 billion parameters to showcase emergent abilities, which is not the case here.

***

**References**

1. Voice2series: Reprogramming acoustic models for time series classification, ICML 21
2. Wavprompt: Towards few-shot spoken language understanding with frozen language models, Interspeech 22
3. Comparison of Multilingual Self-Supervised and Weakly-Supervised Speech Pre-Training for Adaptation to Unseen Languages, Interspeech 23
4. Parameter-Efficient Learning for Text-to-Speech Accent Adaptation, Interspeech 23

**Questions:**

- high level question
   - the authors mainly highlight their claims on the perspective of foundation acoustic model adaptation, only a single pre-trained model has been discovered. A better takeaway to the community could be how different pre-trained speech and acoustic models in response to the input based adaptation.
    - If the authors could provide studies different pre-trained acoustic models (e.g., AudioSet pre-trained, WavLM, Whisper, joint supervised and unsupervised w2v2 based methods) to have a deeper look on the (a) training data source and (b) size of model such as [5], this work could be with larger impacts.
  - how's the proposed methods different with reprogramming and jointly reprogramming plus adapter [6] in the prior works?

- low-level questions
  - what is the trainable parameters in both settings?
  - in terms of acoustic modeling, any acoustic classification or speaker-level open set has been studied in the proposed method?
  - I am curious how the entropy based additional losses training gonna impact the latent space distance measured from w2v2 pre-trained to target open set


***
**References**

5. How to Estimate Model Transferability of Pre-Trained Speech Models? Interspeech 23
6. From English to More Languages: Parameter-Efficient Model Reprogramming for Cross-Lingual Speech Recognition, ICASSP 23


## post-discussion

- I raise my score to six (not argue to reject) and give conditional accept to this work toward checking their statements on
   - including frozen model adaptation discussion
   - Bit and P-tuning with revised entropy loss training
   - total trainable parameter numbers

in their final version. Flagged for AC and SAC double checking if the recommendation is accepted.

**Details Of Ethics Concerns:**

no major ethic concerns.

---

> ### Author Response · Authors · 2023-11-20
>
> Thanks for the careful review and constructive suggestions. Below are our detailed responses to your questions:
>
> ---
>
> **Q1**: "writing quality could be very improved"
>
> **A1**: We appreciate your feedback regarding the writing quality. We have carefully revised the citation formats and typos in the new version of our paper.
>
> ---
>
> **Q2**: "The related speech processing references are missing", "A well-known recent insight involves latent space alignment via Wasserstein measurements, yet the related discussion is absent here"
>
> **A2**: Thanks for highlighting these references. We have incorporated a discussion of them within the related work section in the revised version.
>
> ---
>
> **Q3**: "how's the proposed methods different with reprogramming and jointly reprogramming plus adapter in the prior works?"
>
> **A3**: In summary, the key difference is that our proposed method performs **unsupervised** adaptation-based entropy minimization while reprogramming and jointly reprogramming plus adapter perform **supervised** optimization with access to annotated target data pairs. Furthermore, while reprogramming-based methods focus on training additional small parameter sets for data input parts or adapters within the model, our method focuses on the feature modulation on norm layers without additional parameter introduction.
>
> ---
>
> **Q4**: "...how different pre-trained speech and acoustic models...", "provide studies different pre-trained acoustic models to have a deeper look", "...only Wav2Vec 2.0 Large has been evaluated. The term "foundation model" is often used for models with over 1 billion or even 100 billion parameters to showcase emergent abilities, which is not the case here"
>
> **A4**: Thanks for the good suggestion. We choose the Wav2vec2 models due to their popularity in the community and we acknowledge the importance of evaluating diverse acoustic models. Hence, we conduct additional experiments using, Hubert-Base, Hubert-Large, WavLM-Base, and WavLM-Large to compare the adaptation performance under different model sizes, and training data sources. The experimental results are discussed in Appendix A.5 in the new version.
>
> Furthermore, we believe that there hasn’t been a clear definition for the "acoustic foundation models" in terms of model sizes. This perspective -- "The term "foundation model" is often used for models with over 1 billion or even 100 billion parameters" -- is often discussed in the context of large language models. We refer to "acoustic foundation models" as those large acoustic models that serve fundamental roles and can be extended to various downstream speech tasks.
>
> ---
>
> **Q5**: "Similar parameter-efficient baselines work on frozen pre-trained adaptation diagrams, making it difficult to justify the novelty on applying trainable layer norms"
>
> **A5**: We hope to kindly clarify that parameter efficiency and applying trainable layer norms are not our main contributions even though our method is indeed parameter-efficient. Rather, we investigate TTA within the audio modality and discover that non-silent frames with high entropy contain vital semantic content that cannot be dropped. This motivates us to develop a learning-based adaptation approach to address it.
>
> Besides, there is a key difference in the experimental setting. That is, the mentioned parameter-efficient baselines are trained using supervised signals, while our method performs unsupervised optimization.
>
> ---
>
> **Q6**: "what is the trainable parameters in both settings?"
>
> **A6**: In the CEA, trainable parameters include affine parameters associated with layer normalization and the feature extractor. In the STCR, trainable parameters are affine parameters associated with layer normalization in the model, as illustrated in Figure 2.
>
> ---
>
> **Q7**: "in terms of acoustic modeling, any acoustic classification or speaker-level open set has been studied in the proposed method?"
>
> **A7**: Thanks for this good suggestion. We mainly evaluate ASR in our work since it is one of the most typical problems in speech processing. We definitely agree that acoustic classification and speaker-related tasks are also interesting and could be extendable tasks in future work.
>
>
> ---
>
> **Q8**: "I am curious how the entropy based additional losses training gonna impact the latent space distance measured from w2v2 pre-trained to target open set"
>
> **A8**: We believe such latent space distance might increase as entropy-based additional losses decrease. Basically, it might not be meaningful for a direct comparison of such distances as we cannot correlate larger distances to better adaptation performance.  Instead, it might make sense for the distance comparison among different adaptation methods. This might be an interesting future work.
>
> ---
>
> We hope these responses comprehensively address your queries, and we are open to any further discussion or feedback.

---

> ### Comment · Reviewer_hWCQ · 2023-11-22
> **Re: Layer norm is associated with trainable parameters**
>
> Thanks for authors for the responses.
> - I am happy to see the paper draft format (citations latex issues and model size issues) and quality have both improved largely.
>
> Besides, there are issues I confirm the authors would have replied with imprecise information. I provide below for further references / help for improvements.
>
> 1. Does layer norm / feature modulation introduce new additional associated with trainable parameters?
>
> - Actually, Layer Normalization (Layer Norm) does have trainable parameters. Specifically, it has two parameters: the scale (denoted as $\gamma$) and the shift (denoted as $\beta$). As the cases in this work, applying an additional layer(s) of Layer Normalization to a frozen model can be considered as introducing new parameters for model adaptation.
>
> This extra norms provides a way to adapt the model to new (open domain) data or tasks without altering the original, pre-trained parameters of the model, which is a strategy often used in transfer learning and fine-tuning scenarios.
>
> As a frozen model adaptation study, the additional loss based training upon the additional trainable parameter is standard for having adapters baseline as the reviewer’s initial review suggestion.
>
> Also, the authors’ response of no extra parameter introduction is imprecise (e.g., difference between acoustic model reprogramming and input prompting).
>
> Please consider to correct it since it will potentially mislead further readers.
>
> 2. The wav prompt reference
>
> - The wav prompt reference is not a rigorous format of in-context learning, where ICL is required sequence stacking in previous rounds forward propagations. Referring as prompting learning is much formal.
>
>
> 3. Language usage of foundation models
>
> - I disagree with the authors’ viewpoint on non large scale (1B+) statement. General purpose pre-trained model and foundation model in terms of scaling effects benefited representation learning are different, since many frozen model adaptation characteristics are highly related to scales of model and data. This point does not affect my final score.
>
> In short, this is borderline paper for me. The authors have made largely improvements from a clear reject initial draft. In the latest draft, there is still a gap in terms of deeper understanding of parameter efficient adaptation that missed other multi-loss based parameter efficient adaptation analysis.
>
> I am ok if the paper got accepted but I would keep my recommendation as borderline reject.

---

> > ### Author Response · Authors · 2023-11-23
> > **Thanks for your feedback**
> >
> > Dear Reviewer hWCQ,
> >
> > Thanks for your response and for recognizing the improvement in the format and quality of our paper by “format and quality have both improved largely”. We sincerely appreciate your valuable efforts in reviewing our paper and would like to address your further concerns below:
> >
> > 1. Does layer norm/feature modulation introduce new additional trainable parameters？
> > - In Sec.4.2 of our paper, we discuss the affine parameters (specifically the scale \gamma and the shift \beta) associated with layer normalization. While we agree that “applying an additional layer of layer normalization can be considered as introducing new parameters”, it is important to note that our method **does not add any new layers of normalization**. Instead, we adapt the affine parameters (the scale \gamma and the shift \beta) of the **existing** layer normalization from the pre-training phase, which means no new trainable parameters are introduced.
> >
> > - Our method **aligns** with existing TTA methods [1,2] in the vision domain, which also do not introduce additional trainable parameters (no additional layer), even though most works in the vision domain adapt the affine parameters of **batch** normalization. Adapting the affine parameters of the normalization layers is a common implementation in TTA, and therefore, we **do not claim the parameter efficiency as our main contribution**.
> >
> > - We would also like to kindly clarify that our TTA-based method performs source-free **unsupervised** adaptation while existing reprogramming or reprogramming with adapters performs **supervised** adaptation. Our primary objective is to address open-world acoustic data shifts, rather than task adaptation.
> >
> > 2. Regarding wave prompt reference [3]
> > - We have the perspective that the referenced work can be interpreted as either in-context learning or broadly, prompt learning. Even though the referenced work is not directly related to our work, we have revised it in the latest version to avoid further confusion.
> >
> > 3. Regarding the language usage of foundation models
> > - Thanks for sharing your great viewpoint. We are open to various perspectives regarding this since there is no clear definition in the acoustic foundation models.
> >
> > Thank you once again for your patience, we are grateful for your guidance and would welcome any further discussions or suggestions you might have.
> >
> >
> > **References**
> >
> > [1] Wang, D., Shelhamer, E., Liu, S., Olshausen, B., & Darrell, T. (2021). Tent: Fully test-time adaptation by entropy minimization. ICLR.
> >
> > [2] Niu, S., Wu, J., Zhang, Y., Wen, Z., Chen, Y., Zhao, P., & Tan, M. (2023). Towards stable test-time adaptation in dynamic wild world. ICLR.
> >
> > [3] Gao, H., Ni, J., Qian, K., Zhang, Y., Chang, S., & Hasegawa-Johnson, M. (2022). Wavprompt: Towards few-shot spoken language understanding with frozen language models. Interspeech.

---

> ### Comment · Reviewer_hWCQ · 2023-11-23
> **Re: partial layer norm training setting**
>
> Thanks for the authors clarified the layer norm has been parts of the frozen pre-trained model instead of adding new layer norms. My original interpretation is indeed with this confusion.
>
> Meanwhile, the partially released setting of layer norm / feature modulation is related to bias-only tuning and P-tuning in parameter efficient adaptation.
>
> In recent understandings, parameter efficient learning (e.g., like this partially frozen adaptation work) performing better in few-shot and low-resource adaptation is due to the generalization loss over universal approximation without modifying the decision boundaries. The data efficiency is not the main focus on the performance-driven parameter efficient learning. I would like to point out that the connection between existing frozen representation learning is important here.
>
> P-tuning in Q, V would often take longer time to process experiments. If the authors would add some preliminary results on their additional losses training works for bias-only tuning (or commit to add it in their final version), I would consider the disconnection of equivalent techniques have been satisfactorily resolved to increase my score to six.
>
> The aforementioned Chen et al. [5] and A. Pasad et al. ASRU 21 have discussed this relationship. The author may consider to add some related large pre-trained AM theoretical findings discussion to have broader impacts in their final version.
>
> Again, thanks the author for the replies. I would avoid to use foundation model for rigor definition and encourage the authors spend some time reading the original foundation model paper.

---

> > ### Author Response · Authors · 2023-11-23
> > **Thanks for the further feedback**
> >
> > We sincerely appreciate the further feedback and suggestions. We make the following promises in the final version:
> >
> > - add preliminary results for bias-only tuning and make the connection between our work with existing parameter-efficient techniques
> > - add a discussion on related theoretical findings of the large pre-trained AMs
> >
> > Furthermore, we will carefully consider the usage of “foundation models” again, and the change of colors for Figure 2.

---

> ### Comment · Reviewer_hWCQ · 2023-11-23
> **Re: minor Figure 2 suggestion**
>
> The authors could consider to put trainable color blocks into red and frozen colors blocks into blue / gray for better visualization toward ICLR audiences.

---

### Official Review · Reviewer_UNGc · 2023-11-01

**Soundness:** 2 fair
**Presentation:** 2 fair
**Contribution:** 2 fair
**Rating:** 5
**Confidence:** 3

**Summary:**

This paper proposes a new approach to test-time adaptation.
Unlike previous work, this work explicitly focuses on sequential data (speech), as opposed to static and independent objects such as images.
The proposed method is a heuristic-free, learning-based, confidence enhanced approach to adaptation.
The authors demonstrate the effectiveness of their method and superiority over existing approaches on synthetic and real-world data.

**Strengths:**

* This paper addresses test-time adaptation for sequential data (speech) and not only for static objects (images).
* The topic is relevant.
* The paper proposes a fully learnable approach for test-time adaptation.
* The experiments show small but consistent gains (up to a few % relative).

**Weaknesses:**

* One of the paper's goals is heuristic-free test-time adaptation. As for me, the authors mainly move existing heuristics into loss functions, which makes it a learnable approach and gets rid of hyper-parameters that need to be manually set (at least when ignoring optimization hyperparameters such as learning rate, etc.). But the heuristics remain, don't they?
* Unlike previous work, this paper focuses on sequential data (here, speech) rather than static objects like images. I commend the authors for this. However, the only twist that the authors add is an auxiliary loss based on some heuristic for speech signals' short-term consistency. According to Table 5, the benefits from this loss term are modest. Is this all we can/need to do for sequential data? See also question 1 below.

* I think there are a few issues with the equations:
  - Eq.(3) & Eq.(4): Are these frame-level (what the text says) or sequence-level (what the equations look like) quantities?
  - Eq.(4): Please fix subscript - subscript S with i or sum over i? Utterance-level E?
  - Eq.(5): Cardinalities of z’ don’t match.
* The text says "The classification of frames as silent or non-silent was determined based on pseudo labels derived from model predictions." For Figure 3, is the classification done once on the initial, non-adapted model or repeatedly on the respective adapted model?
* How are the hyper-parameters (e.g., loss coefficients) selected?
* Could you please clarify what LS-C stands for: LibriSpeech Noise Corrupted (Figure 3) vs synthetic dataset (Sec. 5.1)?
* Something is off with the inline citations.
* Figure 3: Is percentage in [0,1] or [0,100]?
* Typo: "model training _phrase_"

**Questions:**

1. Why do we need the auxiliary loss in Eq.(5) to improve the consistency of speech signals? Could we get rid of this heuristic by using a proper sequence-level loss (e.g., sequence-level instead of frame-level entropy in Eq.(2)) ?

2. Summary of Table 2, for example: 'Ours' outperforms 'SUTA' 0.2 - 1.1% absolute, but WER for lower SNRs is still a multiple of WER on "clean" data (as much as 82.2 vs 17.5). This makes me wonder what the lowest achievable WER (e.g., human WER) is, in particular on the heavily corrupted data sets? This information might be useful to understand the empirical results:

    (i) If bound by information loss of corruption: The proposed approach may be optimal and the small gains highly significant. But the task may be not the best to evaluate the proposed method.

    (ii) If not bound: What is missing to considerably close the WER gap?

---

> ### Author Response · Authors · 2023-11-20
>
> Thank you for your meticulous review and constructive feedback. Below are our responses to your queries:
>
> ---
>
> **Q1**: "One of the paper's goals is heuristic-free test-time adaptation. As for me, the authors mainly move existing heuristics into loss functions, which makes it a learnable approach and gets rid of hyperparameters that need to be manually set (at least when ignoring optimization hyperparameters such as learning rate, etc.). But the heuristics remain, don't they?"
>
> **A1**: Thanks for this good question. We acknowledge that the heuristics still remain. Our "heuristic-free" refers to removing the filter-based method for unreliable data with high entropy. Our original intention is to make this heuristic-based method into a learning-based method. We will clarify our description and revise it in the new version.
>
> ---
>
> **Q2**: "... However, the only twist that the authors add is an auxiliary loss based on some heuristic for speech signals' short-term consistency ...", "Why do we need the auxiliary loss in Eq.(5) to improve the consistency of speech signals?"
>
> **A2**: The rationale behind the short-term consistency regularization is grounded in the inherent characteristics of speech signals. Typically, neighboring speech frames often correspond to the same phoneme or character. As such, this consistency regularization aids in fostering reliable predictions and maintaining coherence in speech frames.
>
> ---
>
> **Q3**: "...Is this all we can/need to do for sequential data?", "Could we get rid of this heuristic by using a proper sequence-level loss (e.g., sequence-level instead of frame-level entropy in Eq.(2))?"
>
> **A3**: Thanks for the great suggestion.  We agree that adopting the sequence-level loss, such as sequence discriminative training loss might further improve the TTA performance. We consider exploring this as a future work.
>
> ---
>
> **Q4**: "a few issues with the equations"
>
> **A4**: We are sorry for the confusion. Eq.(3) and Eq.(4) should be frame-level quantities. We have revised Equations (3) and (4), as well as the cardinalities in Equation (5) in the updated version of our paper. Thanks for pointing out these issues.
>
> ---
>
> **Q5**: "For Figure 3, is the classification done once on the initial, non-adapted model or repeatedly on the respective adapted model?"
>
> **A5**: The classification is done repeatedly on the respective adapted model.
>
> ---
>
> **Q6**: "How are the hyper-parameters (e.g., loss coefficients) selected?"
>
> **A6**: Hyper-parameters are selected using standard search techniques. The implementation details of hyper-parameters can be found in Appendix A.2.
>
> ---
>
> **Q7**: "Could you please clarify what LS-C stands for: LibriSpeech Noise Corrupted (Figure 3) vs synthetic dataset (Sec. 5.1)?"
>
> **A7**: We are sorry for the confusion. LS-C stands for LibriSpeech test-other set Corrupted by Gaussian noises, as initially mentioned in Sec. 5.1. We have made appropriate revisions in Sec. 5.1 for greater clarity.
>
> ---
>
> **Q8**: "Figure 3: Is percentage in [0,1] or [0,100]?"
>
> **A8**: The percentage value is in the range [0,1].
>
> ---
>
> **Q9**: "Something is off with the inline citations." "Typo: "model training phrase""
>
> **A9**: Thanks for pointing out these issues. We have revised them in the new version.
>
> ---
>
> **Q10**: "...This makes me wonder what the lowest achievable WER (e.g., human WER) is, in particular on the heavily corrupted data sets? This information might be useful to understand the empirical results."
>
> **A10**: This is a good question. We agree that the human WER bound can indeed assist in understanding the experimental results. However, it is difficult for us to obtain such numbers as it requires conducting human subjective studies.  Besides, heavily corrupted speech data may not be common cases in the real world. Our primary aim in this experimental setting is to evaluate the trend of model robustness under varying noise levels, and our results demonstrate that our proposed method outperforms all baselines in this regard.
>
> ---
>
> We hope these responses adequately address your concerns and we are open to any further feedback or discussions.

---

> > ### Author Response · Authors · 2023-11-23
> > **Thanks so much for further feedback**
> >
> > Dear Reviewer UNGc,
> >
> > We sincerely appreciate your valuable efforts in reviewing our paper. It would be helpful if you have further feedback or suggestions on our rebuttal. If your concerns have been addressed, would you please consider updating your score?
> >
> > Best Regards,
> >
> > The Authors

---

> > ### Comment · Reviewer_UNGc · 2023-11-23
> >
> > Thank you for the responses. I recognize the small but systematic gains by the proposed method and the improved clarity of writing.
> >
> > > A10: This is a good question. We agree that the human WER bound can indeed assist in understanding the experimental results. However, it is difficult for us to obtain such numbers as it requires conducting human subjective studies.
> >
> > A minimal, low-effort solution might be to listen to a number of utterances and convince yourself (and the reader) that humans recognize the clean and heavily corrupted utterances with similar accuracy.
> >
> > > Besides, heavily corrupted speech data may not be common cases in the real world.
> >
> > Okay. If this is the case, I suggest adding a brief comment explaining the artificial nature of the data.
> >
> > > Our primary aim in this experimental setting is to evaluate the trend of model robustness under varying noise levels, and our results demonstrate that our proposed method outperforms all baselines in this regard.
> >
> > Agreed. But in my opinion, the quality and contribution of this paper largely depends on a careful choice of the experimental setup and a solid/extensive evaluation/analysis.
> >
> > I personally think the paper in its current state is a weak reject, but I'm happy to be overruled by the other reviewers and see the work accepted.

---

> > > ### Author Response · Authors · 2023-11-23
> > > **Thanks for the feedback**
> > >
> > > Dear Reviewer UNGc,
> > >
> > > Thanks for your response and constructive suggestion. We have added a brief comment explaining the artificial nature of the data in **Datasets** of Sec.5.1 in the latest version. We would welcome any further suggestions or clarifications you might have.
> > >
> > > Best Regards,
> > >
> > > The Authors

---

### Official Review · Reviewer_SG39 · 2023-11-01

**Soundness:** 3 good
**Presentation:** 3 good
**Contribution:** 3 good
**Rating:** 6
**Confidence:** 4

**Summary:**

This paper tackles the issue of test-time adaptation (TTA) for Speech Foundation Models, including wav2vec2, hubert, and the like. Conventional TTA methods for visual modality typically aim to minimize label entropy in test utterances. These utterances are chosen based on predefined entropy thresholds to prevent model updates that may lead to mode collapse (Niu et al., 2023).

However, applying this traditional approach to speech models doesn't yield the desired results because non-silent frames in speech models often exhibit high entropy and carry valuable information (thus, they shouldn't be pruned). To address this, the paper introduces a confidence-enhanced entropy method that proves effective for speech models.

Empirical findings from experiments on synthetic and real benchmarks underscore the practicality and effectiveness of this proposed approach.

**Strengths:**

1. The paper clearly highlights the reason for previous approaches to not work in audio modality. It make sense that non-silent frames have a high entropy as they are the ones where the non-blank labels are to be predicted.

2. The empirical results show the usefulness of the approach on various experimental protocol.

3. The paper is well-written and easy to follow.

**Weaknesses:**

1. The contributions seems minimal, as the core idea of TTA (Niu et al., 2023 and other related works cited in the paper) are already available in the literature.

**Questions:**

1. In comparison to different vocabulary sizes, such as employing a BPE model with a larger vocabulary, how does this work perform? Would the entropy for non-silent frames significantly increase under such circumstances?

2. Could this approach be adapted to function with a model architecture similar to RNN-Transducer? What specific modifications would be necessary to enable its compatibility with such an architecture?

3. Wav2vec2 features a self-supervised encoder. Would it be a viable alternative to the proposed TTA approach to fine-tune the wav2vec2 encoder using test utterance audio as a baseline? What advantages or disadvantages might this alternative approach present?

4. Could you provide further clarification on the decoding strategies outlined in Section 5.4 of your paper? Specifically, post-adaptation, how do different inference strategies, such as beam search or greedy decoding, impact the inference process, and what are the key differences in their outcomes?

---

> ### Author Response · Authors · 2023-11-20
>
> We appreciate your thoughtful review and valuable suggestions. Our responses to your questions are as follows:
>
> ---
>
> **Q1**: "The contributions seems minimal, as the core idea of TTA are already available in the literature"
>
> **A1**: We recognize the existence of TTA-related works in the literature. However, our work focuses on acoustic modeling, compared to the prevalent focus on vision tasks in previous studies. Specifically, our key contributions include:
> - Identifying that noisy speech frames with high entropy often contain non-silent segments crucial for semantic understanding.
> - Introducing a learning-based adaptation approach, complemented by short-term consistency regularization, to augment the adaptation performance.
>
> ---
>
> **Q2**: "In comparison to different vocabulary sizes, such as employing a BPE model with a larger vocabulary, how does this work perform? Would the entropy for non-silent frames significantly increase under such circumstances?"
>
> **A2**: Thanks for the question. Our proposed method is generalizable to models with large vocabulary sizes. Theoretically, the maximum entropy for non-silent frames is expected to increase due to the larger number of classes. Practically, this might also depend on the test input and models. We conduct an additional experiment using the Conformer-CTC model with BPE tokenization, observing an increase in entropy for non-silent frames from 59.4\% to 70.0\%, as illustrated in Table 2.
>
> Table 2: Entropy Distribution at Step 0 for models with different vocabulary sizes.
>
> | Vocab Size | Wav2vec2 Base | Conformer-CTC |
> | ---    | ---   | ---    |
> | n-sil-h | 0.594 | 0.700 |
> | n-sil-l | 0.406 | 0.300 |
> | sil-h   | 0.362 | 0.497 |
> | sil-l   | 0.638 | 0.503 |
>
>
> ---
>
> **Q3**: "Could this approach be adapted to function with a model architecture similar to RNN-Transducer? What specific modifications would be necessary to enable its compatibility with such an architecture"
>
> **A3**: We think that our proposed approach is adaptable to architectures like RNN-Transducer. The primary modification would involve basing the prediction probability and pseudo labels on the decoder outputs.
>
> ---
>
> **Q4**: "... Would it be a viable alternative to the proposed TTA approach to fine-tune the wav2vec2 encoder using test utterance audio as a baseline? What advantages or disadvantages might this alternative approach present"
>
> **A4**: Thanks for the question. Self-supervised fine-tuning wav2vec2 encoder could serve as a baseline. However, in our experimental setting, we consider a more practical setting where we only have access to a single utterance for adaptation before test-time inference. In this setting, self-supervised fine-tuning wav2vec2 using such a single utterance poses two major disadvantages:
> - The insufficiency of a single test utterance for effective batch construction, leading to data imbalance issues.
> - Potential distortion of the models' distribution upon fine-tuning the entire wav2vec2 encoder.
>
> ---
>
> **Q5**: "Could you provide further clarification on the decoding strategies outlined in Section 5.4 of your paper? Specifically, post-adaptation, how do different inference strategies, such as beam search or greedy decoding, impact the inference process, and what are the key differences in their outcomes?"
>
> **A5**: The performance differences between the two decoding strategies are detailed in Table 4, Sec.5.3.2. Beam search incorporates an additional language model in the decoding process, while greedy search does not. This language model, with its inherent text distribution prior, can help eliminate impossible text combinations during decoding. Moreover, the inclusion of a language model provides a mechanism to address text-domain shifts.
>
> ---
>
> We hope these responses comprehensively address your concerns and we remain open to further discussion and feedback.

---

> > ### Comment · Reviewer_SG39 · 2023-11-23
> >
> > I appreciate the response. I read all the reviews and corresponding rebuttal and I would like to keep my ratings.
> >
> > Just for Information, I want to share a paper that do test-time adaptation via in context-learning (COSMIC: Data Efficient Instruction-tuning For Speech In-Context Learning). Table 6 of the paper might be similar to what you are proposing.

---

> > > ### Author Response · Authors · 2023-11-23
> > > **Thanks for the feedback and sharing**
> > >
> > > Thanks for the response, and for sharing the concurrent and interesting work. This work considers supervised adaptation but offers a cost-effective solution for the adaptation. We will consider discussing it in the final version.

---

### Official Review · Reviewer_cqyf · 2023-11-02

**Soundness:** 3 good
**Presentation:** 3 good
**Contribution:** 3 good
**Rating:** 6
**Confidence:** 2

**Summary:**

The paper investigates Test Time Adaptation of pre-trained acoustic models. different real world shifts like noise, sining voice and accents are examined. For adaptation two issues are considered one is correlation between speech frames and the other is enhancing the confidence. Experimental results supports the gain of the proposed method.

**Strengths:**

- An important issue is addressed
- Paper is written well

**Weaknesses:**

- Experiments could be extended

**Questions:**

- In the ablation study the impact of the two terms in equation 5 is explained. I think the conclusion about the impact of components depends on the setups. The importance of the two components could be different in the noise shift than the accent shift. what do you think?

- Do you consider language as a shift? could a foundation model be adapted to a new language?

- what is a practical way to get alpha in equation 5?

---

> ### Author Response · Authors · 2023-11-20
>
> We are grateful for your insightful feedback and offer our responses as follows:
>
> ---
>
> **Q1**: "...The importance of the two components could be different in the noise shift than the accent shift. what do you think?"
>
> **A1**: Thanks for this question. We agree that the importance of two components may indeed be different in the noise shift and accent shift because these shifts come from distinct sources. Therefore, we conduct additional ablation studies for them respectively. Specifically, we conduct experiments using the dataset with level-3 Gaussian noises in LS-C for noise shift and speech of speaker BWC with Mandarin as L1 in L2-Arctic for accent shift. All experiments are carried out using the Wav2vec2 Base model. The results are detailed in Table 1:
>
> Table 1: Ablation study of core components in the noise shift and accent shift
>
> | Method | Noise Shift | Accent Shift |
> | ---    | ---   | ---    |
> | Ours     | **24.0** | **23.0** |
> | w/o STCR | 25.1 | 23.4 |
> | w/o CEA  | 35.9 | 26.9 |
> | Source | 39.5 | 28.5 |
>
> These experimental findings indicate a pronounced efficacy of our method in mitigating noise shifts as opposed to accent shifts. We conjecture the reason could be that the shift caused by Gaussian noises for each frame is consistent while other shifts such as accent shift could be different within frames.
>
> ---
>
> **Q2**: "Do you consider language as a shift? could a foundation model be adapted to a new language?"
>
> **A2**: Thanks for the interesting question. In our current experimental setting, we mainly address acoustic shifts including Gaussian and environmental noises, accents, and sung speech. We consider the text-domain shifts within the same language, reflecting variations in the linguistic content or context of the speech data. Adapting the model to a new language is an important research problem in multilingual acoustic models. We consider it as future work.
>
> ---
>
> **Q3**: "what is a practical way to get alpha in equation 5?"
>
> **A3**: Alpha represents the weight for short-term consistency regularization and is a hyperparameter in our experiments. We employ standard grid search techniques to search the value ranging from 0.1 to 1 with 0.1 an interval. The details can be found in Appendix A.2.
>
> ---
>
> We hope that these responses satisfactorily address your questions and look forward to any further feedback or suggestions you may have.

---

> > ### Comment · Reviewer_cqyf · 2023-11-22
> >
> > I appreciate the response

---

### Comment · Area_Chair_DJc6 · 2023-11-21
**Reminder to reviewers to participate in the author/reviewer discussion**

Dear reviewers, this is a reminder that the author/reviewer discussion period ends November 22.

This discussion is indeed supposed to be a dialog, so please respond to
the comments from the authors.

AC

---

### Meta-Review · Area_Chair_DJc6 · 2023-12-06

**Metareview:**

## Scientific Claims and Findings
This paper adapts ideas for test-time adaptation, developed in the computer vision community, to speech recognition with large, pre-trained acoustic models such as Wav2Vec2. The key contributions of the paper are (1) minimization of entropy only for frames pseudo-labeled as speech (what the authors call "confidence-enhanced adaptation") and (2) an auxiliary temporal consistency loss that is lower for smoothly-varying signals. The confidence-enhanced adaptation updates both the convolutional feature extractor of Wav2Vec2 and the layer normalization parameters in the Transformer layers, while the temporal consistency loss only updates the layer normalization parameters. Experiments on adaptation to additive Gaussian noise, natural environmental noises, accented speech, and singing show that the proposed approach can improve over some existing baselines. Ablation studies show that both losses contribute to the performance of the proposed TTA method and that the proposed algorithm can work with other models (HuBERT and WavLM).

## Strengths
- The proposed approach yields consistent gains over the baselines.
- The proposed algorithm is clearly described and simple to implement.

## Weaknesses
- The gains over the baselines, most notably SUTA, are quite small.
- From Appendix A.2, it appears that the learning rates are selected by grid search. It is not stated whether this search is done on a per-test basis. If it is, this limits the usefulness of the approach because the practitioner must choose these rates for each adaptation scenario.
- Given that CTC models do not explicitly model silences (they are simply absorbed by the <BLANK> symbol), this paper needs to more clearly explain how the speech/silence pseudolabels are found.
- The paper neglects some closely related prior work which also learns simple affine transformations inside a DNN acoustic model:
  - P. Swietojanski and S. Renals, "Learning hidden unit contributions for unsupervised speaker adaptation of neural network acoustic models," 2014 IEEE Spoken Language Technology Workshop (SLT), South Lake Tahoe, NV, USA, 2014, pp. 171-176, doi: 10.1109/SLT.2014.7078569
  - P. Swietojanski, J. Li and S. Renals, "Learning Hidden Unit Contributions for Unsupervised Acoustic Model Adaptation," in IEEE/ACM Transactions on Audio, Speech, and Language Processing, vol. 24, no. 8, pp. 1450-1463, Aug. 2016, doi: 10.1109/TASLP.2016.2560534
- The temporal consistency loss appears to conflict with the well-known peaky behavior of CTC (see, for example, Figures 1 and 4 in the original CTC paper by Graves, Fernandez, Gomez, and Schmidhuber - https://www.cs.toronto.edu/~graves/icml_2006.pdf). It would be nice if the paper spoke to this issue.

**Justification For Why Not Higher Score:**

My biggest concerns about the paper are (1) the relatively limited novelty of the work and (2) the extremely small improvements over the SUTA baseline. There's not enough here to clear the very high bar for publication in ICLR.

**Justification For Why Not Lower Score:**

N/A

---

### Decision · Program_Chairs · 2024-01-16

Reject